# OPEN QUESTION ANSWERING OVER TABLES AND TEXT

**Wenhu Chen[1] ***, **Ming-Wei Chang[2], Eva Schlinger[2], William Wang[1], William W. Cohen[2]**
[1]University of California, Santa Barbara
[2]Google Research
{wenhuchen, william}@cs.ucsb.edu
{mingweichang, eschling, wcohen}@google.com

## ABSTRACT

In open question answering (QA), the answer to a question is produced by retrieving and then analyzing documents that might contain answers to the question. Most open QA systems have considered only retrieving information from unstructured text. Here we consider for the first time open QA over *both* tabular and textual data and present a new large-scale dataset *Open Table-and-Text Question Answering* (OTT-QA) to evaluate performance on this task[1]. Most questions in OTT-QA require multi-hop inference across tabular data and unstructured text, and the evidence required to answer a question can be distributed in different ways over these two types of input, making evidence retrieval challenging—our baseline model using an iterative retriever and BERT-based reader achieves an exact match score less than 10%. We then propose two novel techniques to address the challenge of retrieving and aggregating evidence for OTT-QA. The first technique is to use "early fusion" to group multiple highly relevant tabular and textual units into a fused block, which provides more context for the retriever to search for. The second technique is to use a cross-block reader to model the cross-dependency between multiple retrieved evidence with global-local sparse attention. Combining these two techniques improves the score significantly, to above 27%.

## 1 INTRODUCTION

Open question answering considers the problem of retrieving documents from a fixed corpus with a *retriever*, and then analyzes retrieved evidence to provide answers to a given question with a *reader*. Prior open question answering systems focused only on retrieving and reading free-form passages or documents. However, a significant amount of real-world information is stored in other forms, such as semi-structured web tables due to its compact representation to aggregate related information. For example, tables are often used to hold large quantities of related facts, especially numeric facts, such as `Career Statistics for Lebron James`. This type of detailed information is found much less frequently in unstructured text. Tables are also commonly used for collections of homogeneous entities or recurring events, like `List of Periodic Comets` or `List of Champions League Winners since 1966`. Hence tabular information serves as an excellent complement to textual data, especially in the open setting. Despite these advantages, no previous studies have exploited the millions of web tables to augment their open QA system.

In this paper, we describe the first study to jointly exploit tables and text for open-domain question answering. For this purpose, we construct a new dataset, Open Table-and-Text Question Answering (OTT-QA). OTT-QA is built on the HybridQA dataset (Chen et al., 2020), and like HybridQA, OTT-QA questions are multi-hop questions which require aggregating information from both tables and text to answer. However, unlike HybridQA, OTT-QA requires the system to *retrieve* relevant tables and text — in contrast, in HybridQA, the ground truth tables and textual passages required for each question are given. To produce OTT-QA's questions, we begin by re-annotating the questions from HybridQA to 'decontextualize' them—i.e., we make questions suitable for the open-domain setting

---

*Part of this work was done during an internship at Google.
[1]Data was released in https://github.com/wenhuchen/OTT-QA by UCSB NLP Group

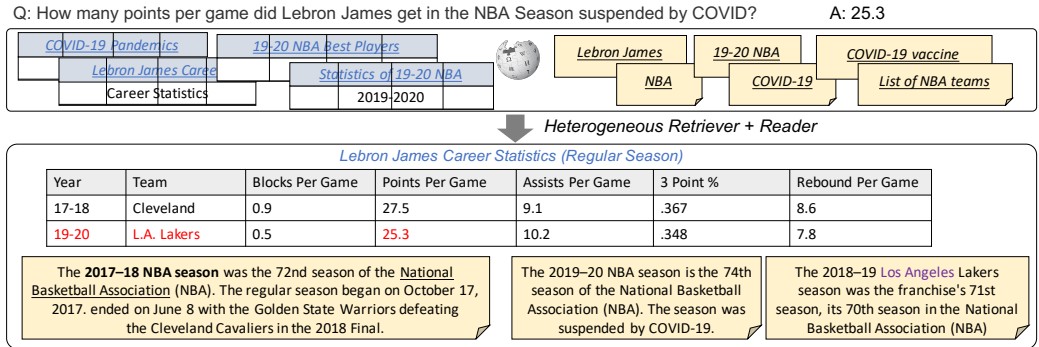

Figure 1: The problem setting: A OTT-QA model needs to retrieve from two candidate pools and then perform multi-hop reasoning to find answers.

so that unique answers can be determined from the question alone, without needing context from the provided text and tables. We then add new questions to remove potential biases. After these steps, OTT-QA contains $45K$ human-annotated questions that require retrieving and aggregating information over tables and text from the whole Wikipedia. Examples from OTT-QA are depicted in Figure 1. Note the table and passages contain non-overlapping information, and both of them must be understood to answer the question. For example, the question has a low lexical overlap with the passage about the 'Lakers', and it needs the table as the bridge to retrieve this passage. Such cross-modality multi-hop retrieval features OTT-QA. More examples are displayed in Appendix.

OTT-QA is distinguished from the existing QA datasets in two aspects. Existing table-based QA datasets (Pasupat & Liang, 2015; Yu et al., 2018; Chen et al., 2020) operates in the closed setting without requiring any retrieval, whereas most existing open QA datasets (Joshi et al., 2017; Yang et al., 2018) require only text retrieval, not table retrieval. One dataset, Natural Questions (NQ) (Kwiatkowski et al., 2019) includes some tabular information in its corpus, but the tables are nearly always of a restricted type (infobox tables with only a single row). In contrast, OTT-QA models require retrieving *both* tabular data and text, and unlike the NQ dataset, requires information fusion from text and tables in non-trivial ways. OTT-QA poses novel and realistic challenges to both the retriever and reader in open QA though the questions are less natural than the real queries from NQ (Kwiatkowski et al., 2019). Retrievers for OTT-QA need to consider two information formats, making the search space larger. Even worse, as questions in OTT-QA often require multi-hop inference, one round of retrieval is often not enough. Readers for OTT-QA also need to aggregate a significant amount of knowledge-intensive information, compared to other reader models: a single table in OTT-QA has an average length of over 300 words. Moreover, readers are often expected to process multiple retrieved units due to the uncertainty in retrieval, which makes it difficult to design strong reader models (Devlin et al., 2019; Liu et al., 2019) with a length limit of 512 tokens.

The baseline system that we propose to address these challenges uses an iterative retriever (Sun et al., 2019; Qi et al., 2019; Min et al., 2019; Ding et al., 2019; Asai et al., 2019) and a BERT reader (Devlin et al., 2019). The iterative retriever explores multiple evidence documents iteratively, interacting with the candidate pool to gradually reformulate the query. Beam search is used to find multiple subsets of documents that may contain all the required evidence, and each subset is then fed to the BERT reader to predict the answer span. The highest-scored prediction is chosen as the answer. The iterative retriever needs to re-encode the query with a big transformer and re-search over the candidate pool, such a procedure (especially dense) can be computationally expensive. Furthermore, the BERT reader fails to capture a global overview of the retrieved documents, which leads to bad local optimum in the model prediction.

We propose a more sophisticated system that addresses these challenges with two novel strategies: namely *fusion* retrieval and *cross-block* reading. The **fusion retriever** first pre-aligns the table segments to their highly related passages, using entity linking. Then, the aligned table segments and passages are grouped as a *fused block*, which contains aggregated information from two modalities; hence, compared to the previous documents, it contains richer context to benefit the following retrieval. We view the fused block as the basic unit to be retrieved, and instead of performing multiple runs of retrieval iteratively, the fusion retriever is used once to retrieve the top $K$ fused blocks; however, due to errors in fusion and retrieval, the retrieved top-1 fused block might not contain

the necessary information. We thus also propose a **cross-block reader** based on a sparse-attention based transformer architecture (Ainslie et al., 2020; Zaheer et al., 2020), which can process extremely long sequences efficiently. We use the cross-block reader to read all the top-K retrieved fused blocks jointly. Both strategies have proven effective compared to the baseline system: the best model combining the two strategies improves the accuracy of the baseline system by a huge margin.

## 2 BACKGROUND

The aim of an open QA system is to extract an answer to a question $q$ from a given large corpus. Most open QA models are retriever-reader models, which extract answers in two steps: retrieval and reading. In the *retrieval* step, a retrieval model $f$ is used to retrieve a set of passages from the text corpus. In the *reading* step, the reader is then used to extract the answer from them.

**Retrieval Function**    There are two commonly-used types of retrieval function $f$: sparse retrievers and dense retrievers. Our sparse retriever uses a unigram-based BM-25 score to retrieve an evidence unit $b$ from the candidate pool $\mathbb{B}$. Our dense retrieval function is a dual-encoder model (Bromley et al., 1994), and we follow (Lee et al., 2019; Guu et al., 2020) for the dual encoder design. The query and the passage are encoded with separate Transformers. As in (Devlin et al., 2019), the vector corresponding to the first token, [CLS], is used as a "pooled" representation of the sequence. The dense retrieval function is the dot product between $h_q = \text{BERT}_{\text{Q}}(q)[\text{CLS}]$ and $h_b = \text{BERT}_{\text{B}}(b)[\text{CLS}]$ for each evidence block $b$ in the candidate corpus—i.e., the scoring function is $f(q, b) = h_q^T h_b$, which can viewed as finding the nearest neighbor in vector space. In the multi-hop open QA setting (Yang et al., 2018), an iterative retrieval function (Sun et al., 2019; Min et al., 2019; Ding et al., 2019) is proposed, which defines the retrieval process as an auto-regressive formula. Our iterative retriever function is denoted as $f([q, b_1, \cdots, b_{j-1}], b_j)$, which appends the previous $j - 1$ rounds of retrieval to the original $q$ in in the $j$-th round of retrieval. Beam search is used in test time.

**Single-Block Reader**    Due to the uncertainty in retrieval, the top-1 document might not always contain the answer. Existing models normally retrieve the top-$k$ documents and feed them to the reader for span selection. The standard reader (Chen et al., 2017; Joshi et al., 2017) aims to extract a span from each of the retrieved blocks $b_i$ and assign a confidence $f(q, b_i)f_{read}(a|q, b_i)$ to it, with $f(q, b_i)$ indicating the retrieval probability and $f_{read}(a|q, b_i)$ denoting the span selection probability by reader. Multiple answers $\{a_1, \cdots, a_k\}$ are ranked with this confidence score and the highest scored answer span $\hat{a}$ is the final answer. Note that the reader needs to run $k$ times, once for each of the top-$k$ retrievals. We refer to this model as the *single-block reader* and use it as our baseline.

**HybridQA**    HybridQA (Chen et al., 2020), a closed-domain QA dataset, is the most related to ours. During the annotation of HybridQA, a table $T$ and its relevant passages $\{P_1, \cdots, P_N\}$ (surrounding text and hyperlinked passage) are given to a crowd worker to write questions which necessarily require both the passage and table information to answer. The original dataset contains 72K multi-hop questions paired with 13K tables with their paired passages. During training/testing time, the ground-truth tables and passages are given to a model, HYBRIDER, to find the final answer. HYBRIDER also serves as an important baseline in our paper.

## 3 TASK AND DATASET

In OTT-QA, the retrieval corpus consists of a set of table candidates $\mathbb{B}_T$ and a set of passage candidates $\mathbb{B}_P$. The task is to answer question $q$ by extracting answer strings from blocks $b \in \mathbb{B}_T \cup \mathbb{B}_P$, where $b$ can be either textual and tabular data. We adopt the standard exact match (EM) and F1 scores (Yang et al., 2018) for evaluation. Different from HybridQA, OTT-QA's table candidates are web tables *without* hyperlinks provided. This decision was made to make the problem setting more general, as otherwise systems that solve OTT-QA could only be applied to high-quality data in Wikipedia. However, in OTT-QA, we provide hyperlinks in the training subset, but not dev/test set. Removing hyperlinks in tables makes the overall task much more challenging, but makes the final systems applicable to more general domains. Thus, an OTT-QA model needs to jointly retrieve both tables and text, without abusing gold hyperlinks, and then aggregate them to find the answer.

**Candidate Pool**    For our table collection $\mathbb{B}_T$, we extracted all Wikipedia regular tables with their metadata including page title, page section title, and section text. The metadata, denoted $T_M$, is essential for de-contextualization. We obtain a table corpus containing over $400k$ high-quality tables

with an average length of 320 words including metadata. For the text passage collection $\mathbb{B}_P$, we crawl English Wikipedia dump pages and filter out noisy pages. We follow HybridQA (Chen et al., 2020) and only keep a maximum of 12 sentences in the introduction section as the passage. We obtain a corpus containing over 5 million passages, with an average of 94 words.

**Notation** We define each table as a matrix $T$, which consists of cells $T_{i,j}$ with $i$ specifying the row, and $j$ specifying the column. Each cell $T_{i,j}$ could be a number, date, phrase or even sentence due to its semi-structured nature. However, a single complete table with structured representation (Herzig et al., 2020) can easily exceed the 512-token limit, which poses great challenges to the downstream reader to process top-$K$ retrieval. Hence we propose to decompose each table $T$ into multiple rows $R_i$, which are combined with the headers, metadata, and global max/min information from the original table as a **table segment**. The table segment is used as the basic retrieval block in our paper. This decomposition procedure increases candidate $\mathbb{B}_T$ from $400k$ to 5 million, making the retrieval problems even more fine-grained and more challenging. Our table segment representation is described in Appendix subsection B.1. In summary, we build a candidate pool of 5 million table segments $\mathbb{B}_T$ and a pool of 5 million passages $\mathbb{B}_P$. We denote as $\mathbb{B}$ as our full candidate pool, which our model needs to find the block $b$ (a table segment or a passage) containing the answer span.

## 3.1 QUESTION AND ANSWER ANNOTATIONS

Our question and answer pairs are built upon the existing HybridQA (Chen et al., 2020) dataset, with several significant changes. First, crowd workers 'decontextualize' the questions so that they are not under-specified or context-dependent, and thus suitable for the open setting. Second, we add more questions to the development/test set to remove possible annotation bias. During annotation, we adopt strict quality control[2] and more details are described in Appendix section A.1.

**Decontextualization** Most questions in HybridQA are contextualized with a given table and several passages, with corresponding questions written by crowd workers. Often, the crowd-sourced questions assume the context. For example, the questions might contain the words `"the players"` because the given table is about `"Netherlands players"`. We thus needed 'de-contextualize' (Parikh et al., 2020) the original context-dependent questions, so they could serve as standalone questions, specific enough to imply a unique answer relative to the corpus. To discourage excessive unwanted modification, we enforce a two-step annotation procedure, as depicted in Figure 2. In the first phase, the worker is only allowed to insert **minimum** words or phrases (or replace pronouns) into the questions based on the information presented by Wikipedia Title, Section Title, and Section Text to make the question have a unique answer. After this step, we often potentially obtain overly-complicated questions that are artificial and unnatural. Therefore, we manually selected the worst 25% questions and sent them back to make them more concise and natural.

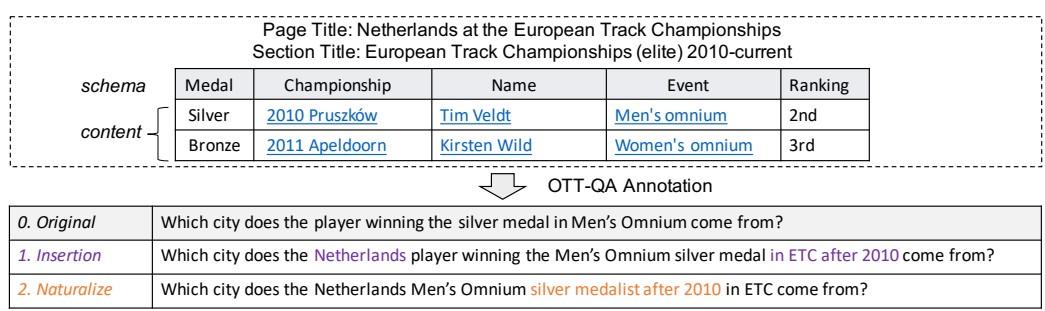

Figure 2: The 'de-contextualization' annotation phase of OTT-QA. In the first step, the annotator is restricted to add phrases from the context. In the second step, the annotator is specifically requested to make the sentence more concise and natural.

**Additional Evaluation Examples** As all the questions from HybridQA are based on the $13k$ tables from the HybridQA set, no questions are asked about the newly crawled $400k$ tables. This

---

[2]The collection is conducted on an established crowdsourcing platform with annotators from countries with English as the native language. The annotators were required to meet the requirements: 1) a native speaker in an English-speaking country 2) having an approval rate of over 95% and 3) having at least 500 approved jobs.

potentially generates unwanted statistical biases or artifacts for the model to exploit, and potentially biases the final evaluation results. Therefore, we randomly sampled another 1100 tables from the newly crawled tables, and follow the original annotation process used by HybridQA to re-collect 2200 new questions. These new questions were mainly used in the dev/test set. Below we refer to the subset of tables used by original HybridQA as the *in-domain* tables.

**Distant Supervision Signals** For the in-domain tables ($\approx 8k$), the cell-wise hyperlinks are provided in OTT-QA as a potential signal for supervision. We use $H_{i,j} = \{b_1, b_2, \dots \in \mathbb{B}_P\}$ to denote the hyperlinks in cell $T_{i,j}$. Since in HybridQA the oracle fine-grained answer span is not explicitly annotated, we approximate this by traversing the table and hyperlinked pasages to find all exact matches. This process contains some noise—a manual study reveals that it roughly contains 15% error. We use this 'weakly-supervised' fine-grained information to train our models. We denote the 'approximate' block of the answer span for answer $a$ as $b_a$, and use it to train our model.

## 3.2 DATASET STATISTICS

After annotation, we sampled roughly 2K questions from the in-domain HybridQA dataset, and then mix them with the newly collected out-domain questions to construct our dev and test sets. Finally, we have 41,469 questions in the training set, 2,214 questions in the dev set, and 2,158 questions in the test set. We conduct more in-detailed analysis over the reasoning types and show them in the Appendix A.3, a remarkable difference from original HybridQA is that a proportion of questions actually have multiple plausible inference chains in the open-domain setting.

## 4 MODEL

Our model for OTT-QA is a retriever-reader model with new designs for both retriever and reader. As discussed briefly above, we propose to use a fusion retriever instead of using a standard iterative retrieve, and we also propose to use cross-block readers to replace a standard single-block reader.

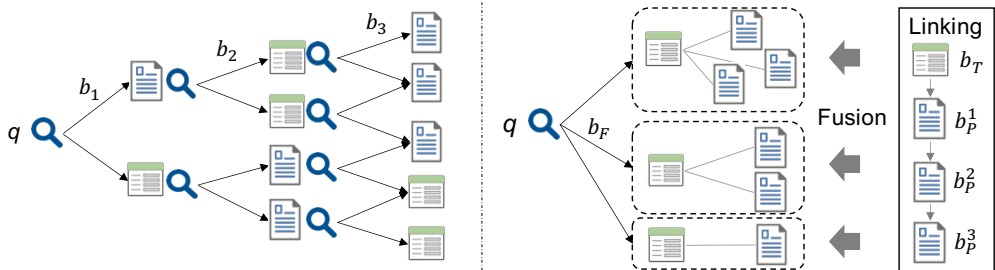

Figure 3: **Left:** Iterative 3-step retrieval over individual blocks (baseline). **Right:** Fusion 1-step retrieval over fused groups, which greatly lowers the cost of iterative encoding and retrieving.

## 4.1 FUSION RETRIEVER

Iterative retrieval (Figure 3, Left) has the following issues. First, iterative retrieval training often requires having supervision signals for every retrieval step to reach good performance, which is not available in OTT-QA. The iterative retrieval also suffers from the problem of error propagation, as early mistakes can propagate to later retrieval stages. Finally, the computation cost for applying a dual-encoder for iterative retrieval is very high, as for every stage, the query embedding has to be re-encoded to include the entire retrieval history.

We propose an alternative strategy to replace multi-step retrieval, namely fusion retrieval (Figure 3, Right). In the fusion retriever, we first use an 'early fusion' strategy to group relevant heterogeneous data before retrieval. The fusion procedure groups several highly-relevant blocks from different modalities as a self-contained group (fused block), which provides more clues for the retriever to utilize. Early fusion is very important for retrieving table segments, which often have incomplete context by themselves. The early fusion process aims to fuse a table segment and relevant pasages into a group. Here we propose to fuse entities mentioned in a table segment to the appropriate passages for those entities; this is similar to document expansion based on a traditional entity linking step. The problem is challenging due to the mismatch between the lexical forms from the

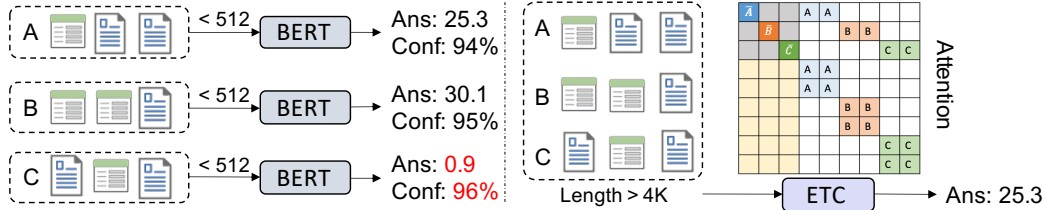

Figure 4: **Left:** Single-block reader with input shorter than 512 tokens (baseline). **Right:** Cross-block reader with length over 4K tokens, and $\bar{A}$ denotes the global state assigned to local block A. The single-block reader is stuck at local optimum, while cross-block reader outputs global optimum.

table (which for brevity are often abbreviated) and the relevant passage titles. For example, a cell in the table of "`NCAA Division I Men's Football Tournament`" contains the term "`Penn State`". Directly matching "`Penn State`" against the passage corpus will lead to "`Penn State University`" rather than the ground-truth hyperlinked entity, named "`Penn State Nittany Lions football`". Therefore, we propose an additional augmentation step, which takes in a table segment block $b_T$ and generates a sequence of augmented queries $q_1, q_2, \cdots, q_n$ token by token to make the queries more similar to the passage title. The augmented queries are then used to search for nearest neighbors in the passage corpus $\mathbb{B}_P$ using BM25 as the final entity linking step, which is depicted in Appendix. The query augmentation is implemented with a GPT-2 model (Radford et al., 2019), fine-tuned on the supervised pairs of (table segment, hyperlink) from the in-domain tables. Each $b_T$ is fed to find its companions $b_P^1, \cdots, b_P^n$, they are collectively called $b_F$.

We follow the standard dual-encoder setting (section 2) and the only difference is that we replace the input of the block encoder with $h_b = \mathtt{BERT_B}([b_T, b_P^1, \cdots, b_P^n])$., which captures the cross-attention between the table and the text within a block. The fused embedding contains richer context from both modalities to complement each other. The retriever only needs to retrieve once from the candidate pool, which dramatically decreases the complexity compared to the existing iterative retrievers.

To enhance the neural retrieval system to retrieve fused blocks, we apply the Inverse Cloze Task (ICT) (Lee et al., 2019) pretraining task on the corpus of fused blocks. ICT is a way to generate pseudo-training data for dense retrieval. Unlike standard document-wise ICT, our fused block contains both table segments and multiple passages. Given a fused block $b_F$, we generate the pseudo-query in the following way: 1) we first corrupt the table segment by randomly dropping half of the words from the table metadata and cells to obtain a partial table segment $\hat{b}_T$. 2) We then randomly sample a sentence $\hat{b}_P$ from the fused passage. We combine $\hat{b}_T$ and $\hat{b}_P$ as a pseudo query $\hat{q}$ and pair it with the original fused block $b_F$ as pre-training data. The pre-training data is applied to enhance the dual encoder's ability to select lexically matched documents. After pre-training, the retriever is fine-tuned on OTT-QA. Finally, at inference time, the retriever is used to retrieve the top $K$ fused blocks for a question, which are then passed to the reader for answer prediction.

## 4.2 Cross-Block Reader

The reader typically needs to process the top-$k$ retrieved blocks returned by the retriever to extract the best answer, as the top-1 block might not contain enough evidence to answer the question. As demonstrated in Figure 4, the cross-block reader aims to address this issue by using cross attention between different blocks to model their dependencies. To obtain the cross-block reader, we take the pre-trained long-range sparse attention transformer (ETC) (Ainslie et al., 2020), which can accept up to 4096 tokens as input, and then fine-tune the model on the distant supervision data. During training, the ground truth (fused) blocks are mixed with hard negative blocks from the retriever. We take the top-$k$ retrieval results to fill the 4096 token space (roughly 15 fused blocks).

Cross-attention between blocks allows a much more powerful way to aggregate information across the $k$ retrieved blocks compared to the single-block reader, especially when the blocks are fused. This is feasible because of the design of the sparse attention structure in ETC, which can constrain the attention of each token to its neighboring tokens within a local radius in its local block. Such

| Retriever | Dev-Sparse | | Dev-Dense | | Test-Best | |
|---|---|---|---|---|---|---|
| Model | EM | F1 | EM | F1 | EM | F1 |
| HYBRIDER (Top-1) (Chen et al., 2020) | 8.7 | 10.9 | 8.9 | 11.3 | 8.4 | 10.6 |
| HYBRIDER (best Top-K) (Chen et al., 2020) | 9.9 | 12.2 | 10.3 | 13.0 | 9.7 | 12.8 |
| Iterative-Retrieval + Single-Block Reader | 9.8 | 13.3 | 7.9 | 11.1 | 9.6 | 13.1 |
| Fusion-Retrieval + Single-Block Reader | 14.3 | 17.8 | 13.8 | 17.2 | 13.4 | 16.9 |
| Iterative-Retrieval + Cross-Block Reader | 17.1 | 20.7 | 14.4 | 18.5 | 16.9 | 20.9 |
| Fusion-Retrieval + Cross-Block Reader | 27.7 | 31.8 | **28.1** | **32.5** | **27.2** | **31.5** |
| † Table-only Retrieval + Cross-Block Reader | 4.6 | 6.9 | 4.9 | 7.2 | 4.4 | 7.0 |
| † Text-only Retrieval + Cross-Block Reader | 8.2 | 12.4 | 8.9 | 12.8 | 8.8 | 12.1 |
| † Oracle Link + Fusion-Retrieval + Cross-Block Reader | 35.8 | 40.1 | 35.2 | 39.9 | 35.0 | 39.5 |
| † Oracle Table + Link (w/o Retrieval) + HYBRIDER | 44.1 | 50.8 | 44.1 | 50.8 | 43.0 | 49.8 |

Table 1: **Main Results**. We conduct experiments with both sparse and dense retrievers using the dev set, and then select the best setting to report the test set results (as indicated by the word "Best"). Fusion-Retrieval and Cross-Block Reader are combined to obtain the highest score. † are ablations.

sparse attention can decrease the attention computation complexity from quadratic $\mathcal{O}(N^2)$ to linear $\mathcal{O}(N|R|)$, where $|R|$ is the local radius (where $N = 4096$ and $|R| = 84$ in our experiments). To allow cross-block interaction, ETC assigns a global state for each local block in the long sequence, and blocks can attention to each other through multiple layers of such global-local structures.

## 5 EXPERIMENTS

All of our code is based on Tensorflow (Abadi et al., 2016). For the retriever part, the sparse retriever is built on top of DrQA (Chen et al., 2017) with unigram features, and the dense retriever is built with BERT. The single-block retriever is based on BERT-uncased, and the cross-block reader is based on ETC (Ainslie et al., 2020). Both of them consist of 12 layers with a hidden size of 768, the minor differences in the relative positional embedding used in ETC. All the models are trained with a learning rate of 1e-5 optimized by AdamW (Loshchilov & Hutter, 2019). We use in-batch negatives (Lee et al., 2019) to train our dense retrievers. A more detailed implementation of the baseline iterative retriever is described in Appendix subsection B.2. In fusion retriever, we use the 'fused' block containing the 'approximate' answer block $b_a$ as the positive instance. In iterative retriever, since the auto-regressive model $f(b_j|q, b_1, \cdots, b_{j-1})$ requires fine-grained inference chain for step-wise supervision, which is not given in OTT-QA. We apply lexical match based heuristics to synthesize inference chains as weakly supervised training data (described in Appendix). For all the dense retrievers, we pre-train with 10K steps using the generated pseudo query and then fine-tune them another 10K step using a batch size of 2048. For the cross-block reader, we fine-tune with a batch size of 64. Both are using 16 cloud TPUs.

**Main Results**   In our experiments, we experiment with different types of retriever and reader models under both sparse and dense setting, the details are described as follows:

● HYBRIDER: this model, designed for closed domain HybridQA questions, is one baseline. Since this model requires a ground truth table with its hyperlinks to do modularized reasoning, we use BM25 to retrieve the most relevant table and passages to reconstruct an 'approximated' input for this model. We experiment with top-1,2,3,4 cases where we use the answer with the highest confidence as the final result. We also directly feed the ground-truth table and hyperlinks to HYBRIDER, which roughly estimates an upper limit of this task.

● Iterative-Retriever (Sparse): We use a 2-step iterative retriever: in the first step, we apply the question to retrieve the top-10 table segments and top-10 passages. In the second step, we use each retrieved table segment to retrieve its related top-5 passages and concatenate each retrieved passage title with the original question to retrieve the top-5 table segments. We merge and calculate the retrieval score of each unique block and rank them by their score. For the single-block reader, we split the retrieved blocks into 512-token chunks and feed them to the BERT reader. For the cross-block reader, we truncate the top 4096 subword tokens and only feed these tokens to reader.

● Iterative-Retriever (Dense): We use a 3-step iterative retriever. In the first step, we encode the question and retrieve the top-8 blocks (either table segment or passage); in the second step, we

concatenate the previous retrieved block and the question to re-encode the query vector to further retrieve top-4 blocks; similarly, the last step retrieves top-2 blocks.

• Fusion Retriever (Iterative): We use a sparse retriever to directly retrieve the top-15 fused blocks based on bag-of-words BM25 score, and then split it into individual table segments and passage blocks. Since passage could be associated with multiple fused blocks, we merge duplicate blocks and use their summed score. Finally, we rank each block based on its merged retrieval score and truncate the first 4096 subword tokens for the next step.

• Fusion Retriever (Iterative): We use a dual-encoder dense retriever to directly retrieve the top-15 fused blocks, and then follow the same procedure as above. Without specifying the dense retriever uses ICT for pre-training by default.

• Fusion Retriever w/o ICT and w/o GPT-2: these two ablation studies are aimed to show the effectiveness of our proposed ICT pre-training and query augmentation.

The main results are presented in Table 1. First, we can observe that best HYBRIDER top-2 can only achieve a comprised exact match of 9.9% while the oracle HYBRIDER can obtain a score of 44%, which reflects the difficulties of the hybrid retrieval in our dataset. We restrain the retriever to only retrieve table and text to answer the questions and report their results in Table 1, even with the strong cross-block reader, the model only obtains 10% EM. These experiments demonstrate the necessity to integrate information from both forms in OTT-QA.

By combining the standard iterative retriever and single-block reader, we can slightly improve the score can to roughly 10%. By replacing the iterative retriever with the proposed sparse fusion retriever, the EM score can reach 14%, a 4.5% absolute improvement. By replacing the single-block with the proposed cross-block reader, the EM score can reach 17% , a 7% absolute improvement. However, by combining the two strategies, the final EM score can reach 28%, with an 18% absolute improvement, which is greater than the sum of individual improvements. The observation suggests the two components can affect each other in a positive way. We conjecture that the fusion retriever is more likely to retrieve mutually-supportive blocks in a group, which makes the multi-hop reasoning across different blocks easier for the following cross-block reader. In comparison, the iterative retriever retrieves isolated table segments and passages separately, which can easily miss out on the bridging evidence for building the complete reasoning chain. Thus, the cross-block reader cannot maximize its advantage in reasoning across blocks.

By removing the ICT pre-training and query augmentation, we observe that the Dev-EM score drops to 24.6%. By removing the GPT-2 query augmentation, the Dev-EM performance drops to 22.1%. These two results indicate the effectiveness of the proposed two strategies. By replacing the predicted hyperlinks with the oracle links, the fusion model performance can increase by 7% EM. This indicates that there is still plenty of room to improve for the table-passage fusion model.

**Linker/Retriever Results** To understand the results more, we evaluate the standalone table-passage entity linking accuracy and retriever recall.

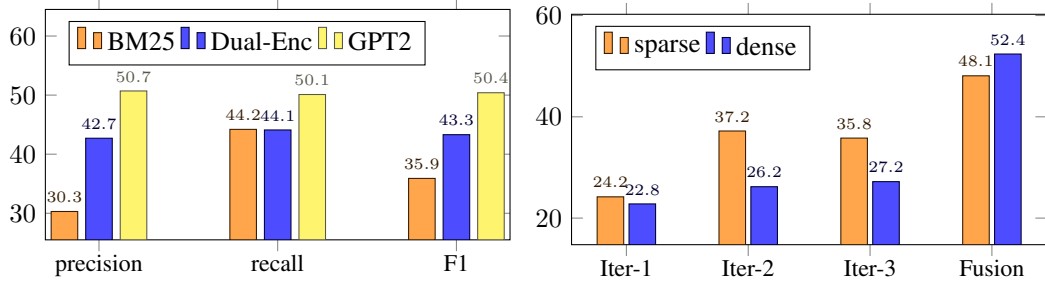

Figure 5: Entity linker performance (F1).    Figure 6: Retriever performance (HITS@4K).

We consider the following linking models: a) BM25 model, which directly uses the cell value to retrieve passages based on their titles without query augmentation, b) a Dual-Encoder model, which encodes the cell value and meta information into a query vector to compute dot-product over all the passage candidate to retrieve, c) a GPT-2 model, which first augments the cell value by the context and then uses BM25. We demonstrate our findings in Figure 5, and evaluate with table-segment-wise

F1 score. We observe that directly using BM25 leads to compromised precision of 30.3%, which is mainly due to the lack of context information. By using a dual-encoder retriever, the precision can be improved to 42%. However, many table segments have either zero or multiple linked passages and can be better modeled by an auto-regressive retrieval process.

We use HITS@4K is used to measure the retriever performance, which indicates the chance of ground truth block existing in the retrieved 4096 subword tokens. The results are reported in Figure 6. We vary the steps of iterative retrievers to show the necessity of multi-hop retrieval in OTT-QA. We observe that the 1-step retrieval has the lowest recall because the answer block in OTT-QA normally has a lower lexical overlap with the query. Adding the second retrieval step can greatly improve the recall, but adding the third retrieval hop has very little impact. In contrast to the iterative retriever, the fusion retriever can consistently improve the performance over the iterative setting for both sparse and dense setting. The sparse setting can rise from 35.8% to 48.1% indicating the advantage of 'early' fusion. The dense retriever's improvement is more dramatic (from 27.2% to 52.4%). We believe this is because the iterative retriever heavily relies on noisy synthetic inference chain data, while the fusion retriever does not require such a fine-grained supervision signal, thus less prone to noise. To better understand the retriever, we conduct detailed error analysis in Appendix C.

## 6 RELATED WORK

**Table Retrieval:** Tables are pervasive on the Web, there have been some studies on mining web tables to answer open-domain questions (Sun et al., 2016; Chakrabarti et al., 2020). In Sun et al. (2016), the authors have proposed a pipeline framework to first detect the topic entity and then generate a candidate chain, finally ranking chains to predict the answer cell. In Chakrabarti et al. (2020), the authors investigate different similarity matching features to retrieve tables from the web. Our paper is significantly different from these two studies in two aspects: 1) the previous papers use private small-scale datasets while we collect a large-scale dataset and release it for public use, 2) the previous studies are restricted to only using tables as evidence, while our paper considers a more realistic and challenging setting with both table and text corpus. Tables have been a ubiquitous information representation form to express semi-structured information. There has been a long-standing effort to utilize tables in natural language processing applications (Pasupat & Liang, 2015; Zhong et al., 2017; Yu et al., 2018; Parikh et al., 2020; Chen et al., 2019). However, these existing tasks are restricted to in-domain cases without requiring any retrieval, and our paper is the first to investigate retrieving web tables for downstream tasks. Another pair of related works are TAPAS (Herzig et al., 2020) and TABERT (Yin et al., 2020), which investigate joint pre-training over textual and tabular data. Our method draws inspiration from these models, and also uses special tokens and embeddings to encode spatial and logical operations inside tables.

**Long Range Transformer:** Recently, many transformer variants to resolve the $\mathcal{O}(n^2)$ attention cost have been proposed including Sparse Attention (Child et al., 2019), Reformer (Kitaev et al., 2020), Routing Transformer (Roy et al., 2020), Longformer (Beltagy et al., 2020) and ETC (Ainslie et al., 2020). These different transformer models apply hierarchical architecture, local-sensitive hashing, global-local state to decrease the attention complexity to nearly linear. Our cross-block reader is based on ETC (Ainslie et al., 2020), but unlike prior works that process one long document for QA, our task requires reading multiple blocks containing both structured and unstructured data. To handle the long sequence of retrieved documents in open-domain question answering, Fusion-in-Decoder (Izacard & Grave, 2020) has been proposed to replace the extractive model with an encoder-decoder generative model. The long sequence of passages are split and encoded independently to decrease the computation complexity, but the decoder still uses full attention over the tens of thousands of encoded vectors to generate the answer token by token. Such full-attention can decrease the decoding speed by an order of magnitude, while our sparse-attention-based cross-block reader can still maintain the same speed as the standard BERT model.

## 7 CONCLUSION

We focus on the problem of performing open question answering over tables and text in this paper. By proposing the fusion retriever and sparse reader, we manage the increase the model's effectiveness and efficiency by a large margin. One interesting question we would like to ask in the future is: can we extend open question answering system to more modalities like images or audios, etc?

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

## A  DATASET COLLECTION

### A.1  DATASET ANNOTATION

**Filtering**  The original HybridQA dataset contains over $72k$ questions paired with $13k$ hyperlinked tables. We adopt two filtering heuristics to make the decontextualization easier. First, we filter out tables without enough meta-information or containing too much non-textual information[3]. Second, we filter out overly-long questions, i.e., questions longer than 30 words. These two filtering heuristics result in a cleaner subset of $46k$ questions paired with $9k$ in-domain tables.

**Quality Control**  During annotation, we conduct strict manual quality evaluation over the decontextualized questions, with the following criteria: 1) the annotated question retains the same semantics and answer as before, 2) the annotated question still requires multi-hop reasoning over both table and passages, and 3) the annotated question is concise and fluent. The manual quality checking was performed over batches distributed to the same annotator. Each batch consists of six questions, one of which will be sampled to decide the acceptance/rejection of the whole batch. The overall acceptance rate for the crowd-sourcing job is 71%, and a rejected job was re-distributed until it was accepted.

### A.2  DATASET EXAMPLES

We demonstrate more examples in Figure 7, which includes more diverse inference chains, like table → text; text → table → text; text + text → comparative → table. Our model is able to perform these reasoning types quite well by jointly matching a query against a fused table-text block.

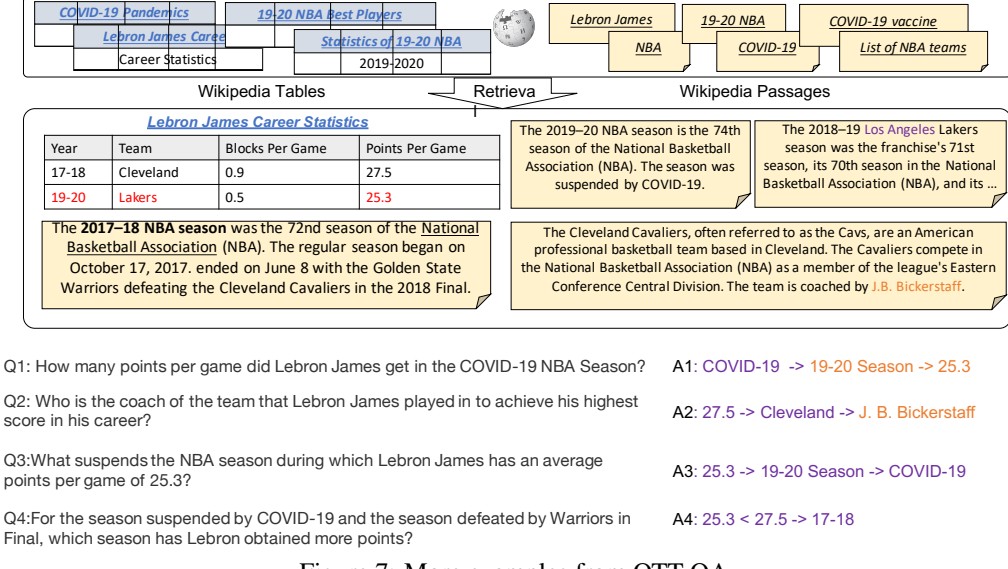

Q1: How many points per game did Lebron James get in the COVID-19 NBA Season?  A1: COVID-19 -> 19-20 Season -> 25.3

Q2: Who is the coach of the team that Lebron James played in to achieve his highest score in his career?  A2: 27.5 -> Cleveland -> J. B. Bickerstaff

Q3:What suspends the NBA season during which Lebron James has an average points per game of 25.3?  A3: 25.3 -> 19-20 Season -> COVID-19

Q4:For the season suspended by COVID-19 and the season defeated by Warriors in Final, which season has Lebron obtained more points?  A4: 25.3 < 27.5 -> 17-18

Figure 7: More examples from OTT-QA

### A.3  QUESTION TYPES

We randomly sampled 100 questions from the dataset to manually analyze the kinds of inference chains seen in OTT-QA and divide the major types into the following categories:

1. Single hop questions (13%) require reading one table or one passage to answer.
2. Two hop questions (57%) require reading one passage and one table to answer. These can be subclassified as 'table bridge' → 'answer text'[4] or 'text bridge' → 'answer table'.

---

[3]Like longitude, latitude, mathematical formulae, etc

[4]I.e., a table forms a bridge between a question and a textual passage, which is read in the first hop, and the answer is extracted from the text.

3. Multi-hop questions (30%) require reading two passages and one table to answer. These mainly following the reasoning chain of 'text bridge' → 'table bridge' → 'answer text'.

4. Questions with multiple reasoning paths: Due to information redundancy in Wikipedia, similar information can appear in both tables and text. We find that 9% of questions are answerable by reading one text passage, 18% of questions are answerable by reading two text passages and 4% of questions are answerable by reading two tables.

## B    MODEL DETAILS

### B.1    RETRIEVAL BLOCK REPRESENTATION

The table decomposition is visualized in Figure 8. The title/section title are prefixed to the table segment. We add the row position token '1st' and a max/min special token over the column to infuse global table information into the segmented unit. The column embedding is added as another vector to the representation. The table segment representation is relatively small and easy to deal with in the following reader model. After the table-passage alignment, we group the highly related

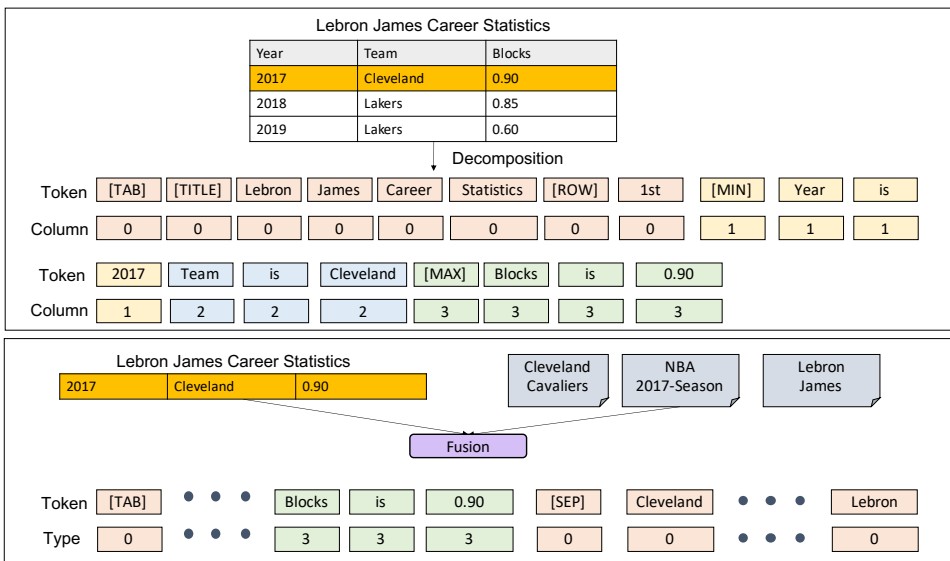

Figure 8: The decomposition of the original table into segments.

units together and represent them as the lower part demonstrated in Figure 8. We add [SEP] tokens to separate different passages and set their type id to 0. Such a flattened representation for fused block $b_F$ will be used throughout our experiments for both sparse/dense retriever and ETC reader.

### B.2    ITERATIVE RETRIEVER

Iterative retrieval has been used in recent graph-based multi-hop retrieval models to gradually retrieve documents to find the correct supporting evidence. Specifically, the retriever conditions the $i$-th round retrieval on the previous round of retrieval results.

**Sparse Retriever**    The sparse retriever uses uni-gram lexical feature to compute the BM-25 score between the $q, .., b_{1..j-1}$ over $b_j \in \mathbb{B}$ to get the top candidates. Here we describe a two-step retrieval procedure, called here the LxM procedure. In the first step, the model calculates the BM25 score between question over all the candidates in $\mathbb{B}$ to select top L/2 table segments $b_T$, and L/2 passages $b_P$. In the second step, the question is concatenated with the retrieved table segment to form L/2 new queries $[q; b_T]$ which are used to retrieve LM/2 passages from $\mathbb{B}$. The question is also concatenated with the retrieved passage titles to form another K/2 queries $[q; b_P]$ to retrieve LM/2 table segments. The retrieval procedure results in at most LM+L unique blocks. Each unique block aggregates its

score from two rounds, denoted as $f(q, b)$, which is used to rank the top-K candidates for the next step. We truncate the top-K candidate by thresholding their combined length.

**Dual-Encoder Retriever** The dual encoder uses a BERT-based encoder to compress each question, table segment, and passage into a fixed-length vector and then computes the dot product between fixed vectors to obtain the highest scored candidates from pool $\mathbb{B}$. However, since the dataset does not provide an explicit supervision signal for the iterative retrieval, we heuristically synthesize some noisy retrieval chains using lexical matching. The retrieval inference chain is depicted as $b_1 \rightarrow b_2 \rightarrow b_K$, which is used to train the model $f(b_k|q, b_{1..k-1})$ in a supervised manner. At inference time, the dual encoder retriever will encode a query $q$ into a fixed vector and retrieve the first $L$ blocks from $\mathbb{B}$. The blocks are appended to query $q$ to form $L$ new queries $[q; b_i]$, which is re-encoded and search for $LM$ new neighbors. We experiment with a maximum of 3-step retrieval of LxMxN to obtain a maximum of L+LxM+LMN unique blocks. Similarly, each unique block aggregates its score from different rounds to select the top-K candidates for the next step.

### B.3 Sparse Fused Retriever

The sparse fused retriever uses the uni-gram lexical feature to compute the BM-25 score between $q$ over $b_F \in \mathbb{B}_F$. The uni-gram feature of $b_F$ is based on the representation depicted in subsection B.1. Note that this BM25 feature will be much more abundant than the BM25 feature in iterative sparse retriever because it encloses more uni-grams. Instead of doing multiple rounds of retrieval, the fused retrieval once retrieve once over the candidate pool and treat all the units inside the block as the same retrieval score. Finally, We truncate the top-K candidate by thresholding their combined length.

### B.4 Query Augmentation

The query augmentation procedure is depicted in Figure 9.

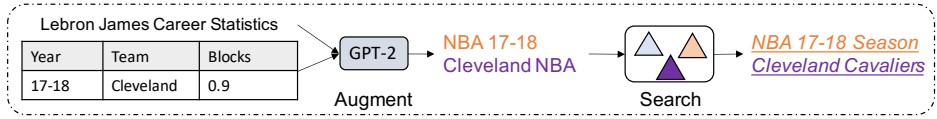

Figure 9: Fusion: 1) GPT-2 query augmentation, 2) nearest neighbor search over passages.

### B.5 Dense Retrieval/In-Batch Negative

Recently, different dense-retrieval methods (Lee et al., 2019; Guu et al., 2020; Karpukhin et al., 2020) based on dual-encoders (Bromley et al., 1994) have been shown to surpass traditional sparse retrieval in open-QA models. The query and the passage are both encoded using a Transformer, which produces a vector for every token. As in (Devlin et al., 2019), the vector corresponding to the first token, `[CLS]`, which is used as a "pooled" representation of the sequence (denoted $\text{BERT}_{\text{CLS}}$). The dense retrieval function can be represented as the dot product between $\text{BERT}_{\text{CLS}}(q)$ and $\text{BERT}_{\text{CLS}}(p)$ for each document in the text collection, much like TF-IDF (Chen et al., 2017) and BM25 (Robertson & Zaragoza, 2009) on some Open QA datasets. To train the dual-encoder, the in-batch negative trick (Yih et al., 2011; Karpukhin et al., 2020) plays an important role, which uses B training instances in each batch and views the other B-1 instances inside the batch as the negatives. In this way, the model reuses computation and effectively trains on $B^2$ question/document pairs in each batch.

## C Performance Analysis

### C.1 Question Type Breakdown performance

We measure our best model's performance (dense fusion retriever + cross-block reader) and baseline model (dense Iterative-Retriever + single-block reader) on different question types (subsection A.3) to show the breakdown statistics in Figure 10 and Figure 11. As we can observe, the gap between

our model vs. baseline in 1-hop question is less significant as 2-hop and 3-hop questions. The iterative retriever's performance is sensitive to the number of hops in the question, which is the largely due to the error propagation in the beam search stage. If the retriever fails to include the golden block in the earlier stage beam, the retrieval in later stage cannot recover from such failure. In contrast, our fusion retriever can group the related information prior to retrieval to retrieve all the blocks at once, which makes the model less prone to the error propagation issue. Another reason is due to the cross-block reader, which can reason over different blocks in the latent space, such implicit reasoning can also decrease the error propagation issue. To sum up, our model is more powerful to deal with complex multi-hop open questions with much less performance drop.

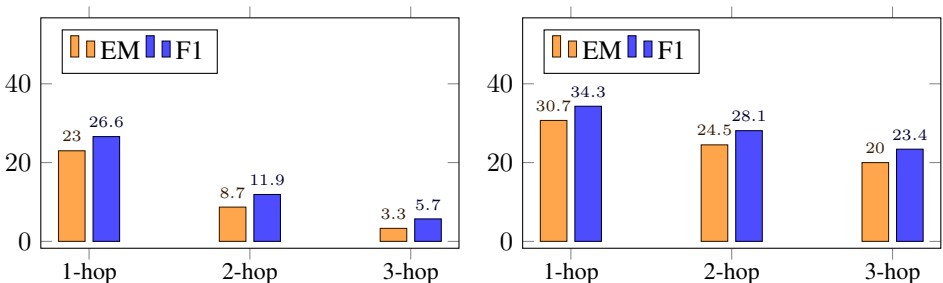

Figure 10: Breakdown for iterative retriever + sing-block reader.

Figure 11: Breakdown for fusion retriever + cross-block reader.

## C.2 RETRIEVER ERROR ANALYSIS

We conduct error analysis to see what are the major issues with the retriever and conclude the following types in Figure 12. The major issues causing the system to retrieve unrelated evidence are low lexical overlap, fusion errors, numerical reasoning and distracting passages or tables. In the

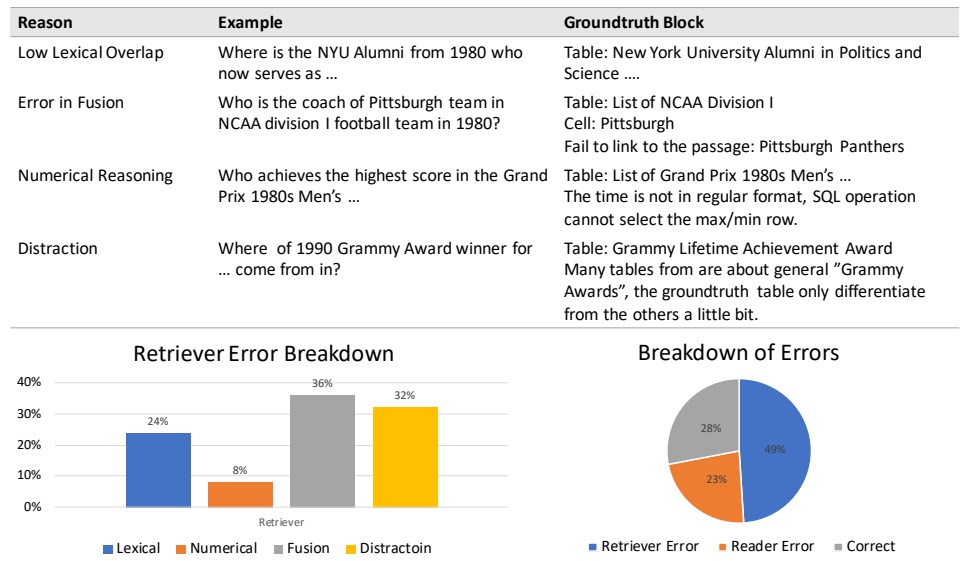

Figure 12: The main error types in the retriever.

Low-Lexical-Overlap case, the errors are mainly coming from the abbreviation, rephrasing of the table metadata, for example, 'New York University' is shortened as 'NYU', etc. In the Fusion-Error case, the issue is mainly because the entity-linking model fails to fuse all the hyperlinked passages, the error (F1=50%) is quantitatively reflected in the entity-linker-performance figure. Numerical-Reasoning error is mainly related to the failure to find max/min/earliest/latest row in the table. The

distraction error is mainly caused by some distracting passages or tables having very similar information. We sample 50 error samples from the dev-set and attribute their errors to the above categories. As shown in the left part, we found that the numerical reasoning error is not as severe as the other three types because the proportion of questions requiring it is relatively small. Besides the low-lexical overlap error, which is general across other open QA datasets like NQ and HoptpotQA, we found the fusion and distraction errors quite specific in our dataset.

- (Fusion) Questions which ask about tables which are linked to too many linked passages. For example, a question over table "Team Record" in `https://en.wikipedia.org/wiki/Sevens_Grand_Prix_Series` is hard because some table rows associate with over 10 passages, it's hard to link them and fuse all of them into a fused block.
- (Distraction) Questions which ask about topics which are contained by too many similar tables, it's hard to differentiate the true one. For example, there are over ten tables in `https://en.wikipedia.org/wiki/List_of_RMIT_University_people`, these similar tables can easily distract the attention of the retriever to select the wrong one from the same page.

From our quantitative results, we can attribute the errors to retriever and reader, among all the examples, 49% of examples cannot find the correct supporting block. For the rest 51% examples with correct block retrieved, the reader fails to select the correct span for 23% of them.

## C.3 LENGTH SENSITIVITY ANALYSIS OF RETRIEVAL/READER

We perform sensitivity analysis for both retriever and reader in Figure 13. We gradually increase

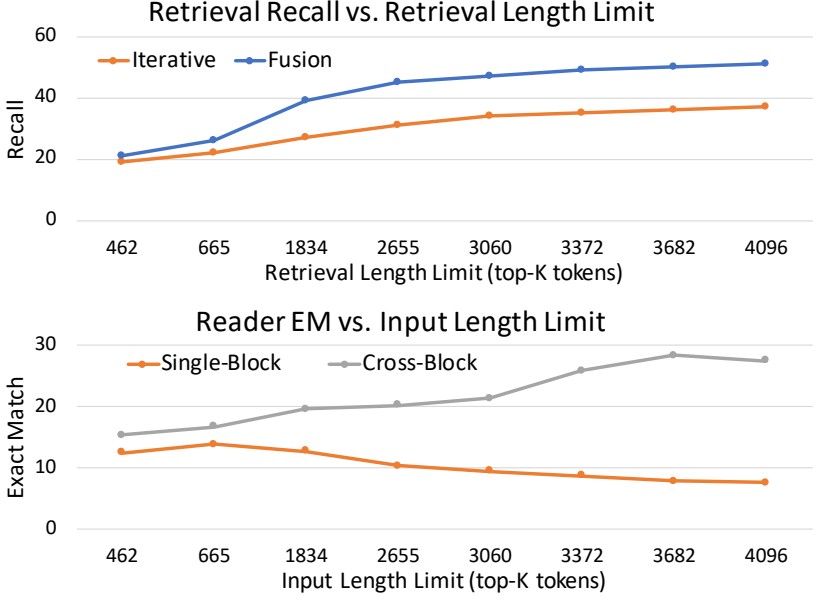

Figure 13: Analyzing retriever performance.

the length limit of retrieved evidence from 400 to 4096 to first visualize its impact on the sparse iterative and dense fusion retriever. For both fusion and iterative retriever, we can observe that both of their recall@K significantly improves as the length limit increases. With a low budget of token limit, their performance is gap is smaller because its performance is dominated by the single-hop questions in the dataset. As the length limit increases, the improvement for fusion retriever is steeper than iterative retriever because the contextualized fusion block becomes easier to retrieve than standalone table segment or passage.

We also visualize the input length's impact on the single-block The performance of single vs cross-block reader. With a low budget of token limit, both single-block and cross-block readers are comparable. However, as the limit increases to 4000, the cross-block reader can digest long input with

its sparse attention mechanism to achieve better scores, while the single-block reader needs to truncate the information to read independently, which leads to a even lower EM score due to introduced noise. This observation reveals the importance of modeling cross-attention between different retrieved evidence units to reach a consistent answer. In single-block reader, dealing with different blocks independently can lead to suboptimal prediction in our dataset.

## D   CONNECTION TO EXISTING WORK

**KB and Text**   The problem combining structured and unstructured data has been studied in question answering. The previous approaches are mainly divided into two categories: 1) FusionNet and PullNet (Sun et al., 2018; 2019) simulate a KB-incomplete setting by masking out some triples from a knowledge graph and use textual information to complete the masked KB triples; these experiments are conducted on KB-based QA datasets. 2) DrKIT (Dhingra et al., 2019) and Knowledge-Guided Retrieval (Min et al., 2019) propose to use entity mentions and relations to guide the retrieval from the web. However, the KB is mainly used as an assisting tool, rather than a necessary information source. In OTT-QA, the structured data is used as necessary information in a realistic setting. The two information forms are combined in a non-trivial way, which makes the problem much harder than the other structure-unstructured QA settings.

**Entity Linking**   Our generative entity linker is related to knowledge-enhanced language understanding (Petroni et al., 2020), which proposes a seq2seq model to deal with different knowledge-intensive tasks like slot filling, entity linking, etc. There is a concurrent related work on auto-regressive entity linking (De Cao et al., 2020), which also demonstrates the advantages of using an autoregressive generation model for entity retrieval.

