# OpenReview forum: "Open Question Answering over Tables and Text"
_ICLR.cc/2021/Conference — ICLR 2021 Poster_

### Official Review · AnonReviewer2 · 2020-10-26
**Good interesting paper but it has some important questions that were not adequately answered**

**Rating:** 6
**Confidence:** 4

**Review:**

-----------------------------
Summary
-----------------------------

In this paper, the authors introduce the task of open domain QA using table and text. Unlike recent tasks on table and text retrieval, where the table associated with the text is known, in this task both tables and text need to be retrieved from a large collection. Similar to the recent HybridQA task, questions may require fusing information both tables and text to answer questions. The dataset has been constructed by reusing the Hybrid QA dataset as well as crawling additional tables and passages from Wikipedia. Since the questions in the dataset may be non-contextual (for instance, not explicitly mentioning the name of a country when the data is only for that country), the authors used crowd-sourced workers to insert words or phrases from the section title, caption etc and make the question self-contained. However, this can cause the questions to become unnatural and excessively long  - the authors manually select the 25% worst questions and get them refined by a second set of annotation tasks. Additionally, to include new samples beyond Hybrid QA, they collect 2200 new QA pairs using the additional tables crawled and include them as part of the dev and test set. The exact answer in the table is annotated using distant supervision and has an extraction accuracy of 85%.  The overall dataset contains 41.5K questions in train and approximately 2K questions each in the dev and test set.

The authors also present a method for for this task - It consists of a "fusion reader" which that aligns a passage and table segments ( Table row+ metadata= Table segment). These are then considered as one unit for retrieval. To identify the passage-segment alignments, it takes a table segment and generates augmented queries to fetch candidate passages.  The augmented queries are generated token-by-token and are used to transform the queries  into the passage titles. This augmented query generator is based on GPT2 and has been fine tuned using the training data. BM25 retrieval is used to fetch passages given the generated queries. Given a table segment and its related (retrieved passages), BERT is used to jointly encode the two sequentially and the CLS token is used to create a representation of the "fused" block.

The retriever uses a question, a fused block and computes the dot product between the BERT CLS embeddings to score blocks (as described in Section 2 but the results section suggests its ORCA -- See Q3). Inverse Cloze pre-training is done on the fused blocks to improve the retriever. Answer generation is done by using the pre-trained long-range sparse attention transformer (ETC) to encode questions and the retrieved blocks to return the answer span.

Experiments have been presented using the HYBRIDER baseline from the HybridQA task as well as an iterative retriever reasoner baseline in place of the fused retriever. Experiments indicate that the use of fusion retrieval as well as cross block attention (using ETC) individually and jointly help improve performance over baselines. It is also interesting to note that the retrieval of fused blocks works better with BM25 (Sparse) as opposed to using BERT.



---------------------------------------------------------
Strengths and Weaknesses
--------------------------------------------------------

Strengths
+  New Task and Model for a challenging problem
+ Reuses existing datasets as well as baselines

Weaknesses

- Paper is a little hard to follow and required multiple readings.  For instance,  the use and introduction of the words such as "block", "segment", "cell", "row" in writing make the paper hard to follow -- perhaps it would be good to define them in the beginning with examples so that it is easy to reference when reading
- Limited qualitative insights and almost no error analysis -- It would have been useful to include an error analysis along with some qualitative insights though the motivation for fused reasoning over iterative reasoning are clear. What could someone do next on this task? The task is far from solved as the results in Table 1 indicate and the information in Figure 9 in the appendix is not adequate.
- Both the iterative reader as well as fused retriever would be sensitive to the value of "top-k" - No study or details have been provided. This is important -- for example, in case of the iterative reader, 50% of the results included are from passages while 50% of the results are from the tables (as per the information in the appendix). Some recall statistics could be helpful (Also see Q2)

---------------------
Questions
----------------------

Q1 What is the performance of models broken down by question types -- the dataset analysis in the appendix suggests two-hop questions dominate the dataset?


Q2 The performance of the HYBRIDER (a model designed for multi-hop inference on text-table data) appears to be lower than the Iterative retriever and single block reader? Could that not be because the BM25 R@1 is likely to be very low -- can the model even do multi-hop reasoning across passages in this case? Perhaps a method that aggregated scores after running HYBRIDER on the top-K retrieved blocks, to return a final ranking may have been a better baseline?


Q3 Unless I have missed it, I'm unclear about the sparse/dense retrieval In Table 1 -- is that BM25 applied on the blocks B_F in raw text i.e table segment + passage? If the dense retrieval is based ORCA what is the text in the second paragraph for? Is that not used for retrieval along with the "retrieval function" in Section 2?


Overall this is a good interesting paper but it has some important questions that were not adequately answered.


------------------------------------------------
Updated after author response
------------------------------------------------
I have updated my overall rating in view of the author response.

---

> ### Author Response · Authors · 2020-11-16
> **Author Response (1/n)**
>
> Notation:
> Thanks for your suggestion, we put all the notation in Section 3, the second paragraph. The most important notation is table segment, which is used throughout the paper.
>
>
> Qualitative insights and error analysis:
> The qualitative and error analysis are added to Appendix C.1 and C.2, we classify questions into 1-hop,2-hop, and 3-hop to show the breakdown performance on these questions. We also classify the errors of retriever into four categories and show the error statistics in these four categories. Besides the general low lexical overlap errors which occur in all previous datasets, our dataset has some unique challenges -- fusion error and distraction error. Fusion error is mainly because a table segment is linked to too many passages (over 10), the system will have a hard time figuring out which passage to hop (like Team Record table in https://en.wikipedia.org/wiki/Sevens_Grand_Prix_Series). Distraction error is mainly caused by too many tables talking about the same topic with very minor differences in meta-data. An example is shown in https://en.wikipedia.org/wiki/List_of_RMIT_University_people, which contains tens of similar tables.
>
>
> “Sensitivity to K and Recall statistics”:
> We perform sensitivity analysis for **both retriever and reader** w.r.t to top-K (K is the token limit) and show our findings in Appendix C.3. We observe that with a low budget of K, iterative and fusion retrievers obtain similar recall performance. As the budget increases, the improvement curve for fusion retriever is steeper than the iterative one. We conjecture that this is because of the more context included in the fusion block, which makes retrieval easier.
> For readers, the single-block and cross-block are both achieving similar scores with tokens within 500. By increasing the budget to 4000, the cross-block reader can digest the long input information to improve the performance, while the single-block reader needs to read multiple times, which greatly restricts its performance gain.
>
> "50% of the results included are from passages while 50%...":
> Not exactly 50% and 50% for table and passage, it’s dependent on the question. Different questions could lead to totally different distributions over the two forms. In the best retriever, the passage/table ratio is roughly 2:1 in the final top-K tokens.
> For sparse retrievers, let’s say we do 8x4 retrieval, we will split 8 into 4 passages and 4 table segments, and the next 4 as its complementary form. After retrieving these 32 + 8=40 blocks, we will merge and rank them by matching scores, so not entirely 50% and 50%, sometimes the table segments are dominating the top-K tokens and sometimes passages will dominate.
> For the dense iterative retriever will retrieve the highest-scored block regardless of its form. As long as the block has a high matching score, the dense retriever will retrieve it.
>
> Q1: Break-down statistics:
> We added break-down statistics in Appendix C.1. We sample questions and manually classify them into 1-hop, 2-hop, and 3-hop questions to show the model's breakdown performance on them. We found 2-hop and 3-hop questions are generally getting a much lower score than 1-hop questions. The 2-hop question is the most typical question in the dataset, the overall performance is mostly affected by this type.
>
> Q2: HYBRIDER with top-k:
> This is a good suggestion, we tried to run HYBRIDER on top-K retrieval and select the one with the highest confidence score in the answer span selection module. The best performance is achieved by top-2, which can gain 1% improvement over top-1. Adding more top-K results can harm the performance due to more noise.

---

> > ### Author Response · Authors · 2020-11-16
> > **Author Response (2/n)**
> >
> >
> > Q3-1: Is that BM25 … table + passage?
> > Yes, we will concatenate the table segment and the passage like two strings and then calculate the BM25 feature based on the concatenated string. Please see Appendix B.1 to see how we encode fusion blocks as if they are standard passages.
> >
> > Q3-2: Is the retrieval function … Section 2?
> > The **retrieval function in Sec 2 is used everywhere**, both for iterative retriever and the fusion retriever. Their architecture and parameters are exactly the same, the only difference lies in the input to $BERT_{b}$. For standard iterative retriever, it encodes a single table segment or a single passage. For fusion retriever, it concatenates the table segment and passages and adds some special tokens like [SEP], [TAB]. [DOC], [MAX], .. to make the representation more informative. Please see Appendix B.1 to see how we encode fusion blocks as if they are standard passages.
> >
> > Q3-2: If dense … is based ORQA …?
> > We **didn’t directly use the ORQA code** because they are pre-trained to retrieve documents rather than fused blocks. We borrow ORQA’s ICT concept to design **Fused ICT** pre-training. Specifically, we extract some n-gram phrases from the table segment (title, cell value, etc) and some phrases from one of the potentially linked passages, and concatenate them as a ‘fake’ query. This ‘fake’ query and its original fused block are used as the synthetic data to pre-train the dual-encoder. This is talked about in the last paragraph of Sec 4.1 in the paper.

---

### Official Review · AnonReviewer3 · 2020-10-28
**Review #3**

**Rating:** 7
**Confidence:** 4

**Review:**

This paper proposes a new setting of open-domain question answering. Usually, we only retrieve question-related text from the web or Wikipedia for answering questions. The authors build up a new dataset which need to retrieve both text and the corresponding table to answer open-domain questions. This setting is more close to a real world setting where structured information is also essential. Moreover, the authors propose a pipeline of fused retrieval and cross-block reader to solve the problem. This baseline is very strong and consists of many SOTA methods such as ICT, ETC. And I like the idea of fused retrieval which is very important to build a connection between table and text. Although the idea is close to the entity linking or hyperlinks for multi-hop QA, it is new under this open QA setting. Overall, I would like this paper accepted.

Pros:
1. Release a new task and dataset for answering open-domain questions with text and table. The dataset is carefully annotated by two steps by making use of decontextualization method.
2. Set up very strong baselines, and propose fused retrieval which is important to achieve strong performance for this task.
3. This paper is well-written and easy to read.

Cons:
1. I would like to see more analysis on the dataset. For example, what's the distribution of different question types? What kinds of questions are hard to solve? Can the proposed model solve reasoning problems from the dataset?
2. Missing baselines of using text only and table only to answer the question. It's unclear whether we do need both text and table.

###update###
I have read the other reviews and the author feedbacks. I would like to keep my rating.

---

> ### Author Response · Authors · 2020-11-16
> **Author Response**
>
> “More Analysis to show breakdown performance”:
> We added multiple error analysis in Appendix C.
> In C.1 we sample questions and manually classify them into 1-hop, 2-hop and 3-hop questions to show the model's breakdown performance on them. We found 2-hop and 3-hop questions are generally getting a much lower score than 1-hop questions.
> In C.2, we classify the retrieval error into more fine-grained categories and show the retriever’s error breakdown in these categories. Besides the well-known low-lexical overlap error, we found that fusion error and distraction error are two major error sources. This analysis shed lights on the “what questions are hard”.
>
> “What questions are hard”
> Well-known type: questions which have low lexical overlap with evidence.
> (Fusion) Questions which ask about tables which are linked to too many linked passages. For example, a question over table “Team Record” in https://en.wikipedia.org/wiki/Sevens_Grand_Prix_Series is hard because some table rows associate with over 10 passages, it’s hard to link them and fuse all of them into a fused block.
> (Distraction) Questions which ask about topics which are contained by too many similar tables, it’s hard to differentiate the true one. For example,  there are over ten tables in https://en.wikipedia.org/wiki/List_of_RMIT_University_people, these similar tables can easily distract the attention of the retriever to select the wrong one.
>
>
> “Missing baselines”:
> Text-only and table-only baselines: We add these two baselines in our new revision in Table 1, both models can only obtain EM scores between 4%-8%, which reflects the necessity to combine information from both sides to answer the questions in OTT-QA.

---

### Official Review · AnonReviewer1 · 2020-10-29
**new dataset to support research on open QA over text and table; two techs to retrieve and aggregate evidence; good empirical results**

**Rating:** 7
**Confidence:** 4

**Review:**

##########################################################################

Summary:


The paper provides a interesting direction in open question answering. In particular, it proposes an open QA problem over both tabular and textual data, and present a new large-scale dataset Open Table-and-Text Question Answering (OTT-QA) to evaluate performance on this task. Two techniques are introduced to address the challenge of retrieving and aggregating evidence for OTT-QA. Results show that the newly introduced techs bring improvements.

##########################################################################

Reasons for score:


Overall, I vote for accepting. I like the idea of open question answering with various types of evidence. The major contribution of this work, in my personal opinion, is the the creation of the dataset which would foster the research on open question answering over text and table. The techs introduced are sound but the novelty in terms of methodology is limited.


##########################################################################Pros:
Comments:

1. The paper formulate an interesting problem of open QA problem over both tabular and textual data.


2. The creation of the dataset (OTT-QA) is a great contribution to the community. The authors claim to release the data to public. Would the test set make blind so that make it a challenge like SQuAD?


3. The method is sound. Experiment study is convincing. Two introduced techs bring improvements.


##########################################################################

---

> ### Author Response · Authors · 2020-11-16
> **Author Response**
>
> Thanks for your comments, there will be a leaderboard for OTT-QA. We hope to use this leaderboard to push the current community to invent more to design universal QA systems that are agnostic to the knowledge form and can be generalizable to the web-scale.
>
> Btw, we add more ablation results to the main table and error analysis in the Appendix to help the reviewer get a better understanding of the dataset and proposed model.

---

### Official Review · AnonReviewer4 · 2020-10-30
**New task, a makeshift dataset and baselines. Less scientific and more engineering.**

**Rating:** 6
**Confidence:** 4

**Review:**

(This review is a collaboration between a junior and a senior reviewer as part of the training to the junior reviewer. Both of them read the paper in detail.)

Summary

This work extends the task of answering questions over tables and text to open-domain. They construct a new dataset - OTT-QA - on top of a closed domain multi-hop question-answering dataset, which requires reasoning over tables and text. Adding a retriever step poses a challenge for the system to retrieve relevant tables and text, given a question. The authors propose two strategies for retrieval - 1) preprocessing the dataset offline to group tables and text into blocks, which are later retrieved by the retriever, 2) using long-range sparse attention transformers to read multiple retrieved blocks at once. Applying both strategies gives significant gains over other baselines.

Strengths

New task and a dataset for this task by converting HybridQA to an open-setting.

New baselines for this task.

Weaknesses

The proposed dataset is a makeshift dataset that is wrapped on top of HybridQA. This makes the task artificial. HybridQA is built by first selecting a table and then related anchoring documents. Whereas in a true open-setting, there may not be such dependence. A question may be answered on document1, which could lead to document 2 and then a table so on (i.e., the table and document1 need not be related).

Apart from the retrieval, no qualitative differences between HybridQA and OTT-QA are presented.

The reviewer is left with distaste making them wonder what are the scientific takeaways from this work. Currently, the paper feels like here is a dataset and here are a bunch of models. What are the unique linguistic phenomena present in this task? Is it mainly the retrieval that is hardest or the reasoning (reading) once retrieved?

Experiments:

1. An oracle baseline where the gold table is given but not the documents (and vice versa) should be presented to understand the impact of HybridQA annotation procedure on the naturality of this task. Due to the annotation biases, the retrieval task could boil down merely to retrieving the correct table (and its associated documents).

2. Pre-training is a computationally expensive operation. As a practitioner who wants to apply or extend the methods proposed in this work, it would be helpful to know how much performance gains are stemming from pre-training of the neural retrieval system.

3. Results using just tables and just documents are absent.

4. The supervised pairs of (table segment, hyperlink) are essential for forming the fused blocks, however, this information cannot be used by iterative retrievers. What will be the performance of the fusion retriever in the absence of hyperlinks? i.e. without using GPT-2 for query augmentation, or using it without fine-tuning. This will quantify the importance of hyperlinks as a supervision signal.

Questions for the Authors:

1. How do you define over-complicated, artificial and unnatural decontextualized questions?

2. The authors say one round of retrieval is not enough for OTT-QA because questions require multihop. This is just an artifact because they split the table into multiple blocks whereas in the actual HybridQA setting, all you require is one table and an associated document(s). Doesn’t this mean only one round of retrieval is just enough if you use full-table and associated documents?

Suggested improvements in paper presentation

Although the paper mentions that a dual-encoder design is used, the current notation can give a wrong impression that a single encoder is generating the representation of the query and the block. I would request the authors to consider using the notation for dual-encoder as used in [1].
Modify the Lebron James Career Statistics table in Figure 1 and Appendix subsection A.5 to denote “L.A. Lakers” instead of “Lakers” as given in the table on Lebron James’s Wikipedia page.
Table segment representation is not very clear from Figure 8 of Appendix subsection A.5. I believe the last 4 tokens should be “[MAX] Blocks is 0.90” instead of “[MAX] Blocks is Cleveland”. It would be helpful to have an example in which the minimum and maximum entries are from rows different from the row being encoded.
It will be helpful to the reader if there is a paragraph in the Appendix containing details of the formulation of the sparse Fusion Retriever.
Include specific Appendix subsection in Section 1 (“ More examples are displayed in Appendix”) and Section 5 (“weakly supervised training data (described in Appendix)”). Remove “discuss in Appendix” from the Dual-Encoder Retriever paragraph in Appendix subsection A.6.

Typos
Therefore, We -> Therefore, we
Table 1 caption  - “both brings significant improvement.” -> “both bring significant improvement.”
Appendix subsection A.1 - “Secone, we filter out” -> “Second, we filter out”

[1] Latent retrieval for weakly supervised open domain question answering, Lee et al., ACL 2019

---

> ### Author Response · Authors · 2020-11-16
> **Author Response (1/n)**
>
> “a makeshift dataset... on top of HybridQA.. makes the task artificial”:
> The specific novel task is retrieving and reasoning over both structured and semi-structured data in an open-QA setting. While we agree that the use of crowdsourcing is somewhat artificial, it is the process used for almost all public QA datasets like SQUAD, WikiHop, HotpotQA, etc (except NQ), as the real user query data is often difficult to access. Though being suboptimal, we optimize the annotation procedure to control the bias and simulate the diversity in real-world search engines. We apply strict quality control to harvest diverse sets of questions, which can reasonably cover the potential reasoning types in real-world applications.
>
> “True open-domain setting, there are … so on”:
> While the mentioned specific type of 3-hop text>text>table questions are not covered in the dataset, our dataset does cover a **wide range of 2-hop, 3-hop questions like table->text, text->table, text->table->text**. Due to information redundancy, there also exist table->table, text->text questions, please refer to Appendix C. for the breakdown statistics of these different types. Though the questions are decontextualized over closed-domain HybridQA, it does **not** necessarily mean the first hop is always retrieving the table. During decontextualization, the workers are required to add **minimum table meta information** to make the answer unique rather than adding **all table meta information**. Thus, some decontextualized questions can be very vague about the table. However, these questions are very specific about an entity (passage), which makes it easier to first retrieve the entity passage and then retrieve the table which has a potential hyperlink to this passage. For example, a minimally decontextualized question could be“ … Olympic Swimming Butterfly … Something about Henrique Martins …?” regarding the table in (https://en.wikipedia.org/wiki/Swimming_at_the_2016_Summer_Olympics_%E2%80%93_Men%27s_100_metre_butterfly). Since “2016” and “heat” are not mentioned, there could be many tables regarding “Olympic Swimming Butterfly”, it’s impossible to retrieve this specific table directly. However, if we first retrieve “Henrique Martins”, since he only participated in one Olympic, we can then narrow down to this specific table. Our observation of sampled 100 questions estimates that over 30% are falling into this type.
>
> “What are the unique linguistic phenomena”:
> It’s linguistically different from these datasets from the following aspects: 1) **more numeral-related questions** are asked, like “How many votes did XXX lead in the 2020 presidential election in ...”. This information is normally non-existed in any text or KB. 2) **the domain is quite different**, the questions from previous datasets are mostly about specific items like people, book, film, etc. Our dataset contains questions about **recurring events in sports, finance**, politics like “Among all the U.S players from Kansas, who won … in XXX game?”. These questions normally require aggregating information over a set of items. Answering these questions is difficult by simply using text because the model might need to retrieve a set of passages from everywhere to conclude the answer. However, tables provide a shortcut to answering these types of questions. These questions barely exist in the previous open-domain text QA dataset, but it’s ubiquitous in OTT-QA. Therefore, we think OTT-QA is a good complement to the current QA datasets.
>
>
> “Takeaway”:
> the takeaway message of the paper is “Open-QA systems should gather and reason with information in diverse formats as it’s common in real-world applications. Experimentally, we have made big steps toward this goal, by developing a novel dataset, and exploring novel techniques that dramatically improve performance over plausible baselines. ”. We do propose a number of new models, but this is because the initial baselines for the task perform so poorly: we wanted to make a case that the problem is not too hard to approach with current methods. To sum up, our paper sheds light on how to integrate web-scale heterogeneous evidence to answer any open-domain questions and our proposed method has low computation complexity, which makes it possible to be applied to real industrial applications.
>
> “Is it mainly the retrieval that is hardest .. once retrieved?”:
> We think the challenges are in two folds, 1) the retrieval is challenging due to the existence of different forms of data, though text retrieval has been extensively studied. The problem of table retrieval is less explored, not to mention the joint retrieval over both forms with mutual dependency. 2) the reasoning challenge is discussed in the original HybridQA, but the close-domain setting only refers to deal with information from one table. In the open-domain setting, the reader is presented with an even larger set of tables and passages, the challenge to deal with the uncertainty is magnified.

---

> > ### Author Response · Authors · 2020-11-16
> > **Author Response (2/n)**
> >
> > “Oracle baseline”:
> > the oracle baseline is demonstrated in Table 1. As can be seen, by replacing the GPT-2 entity linker with the ground truth hyperlink, the performance can improve by 7% from 28% to 35%. By feeding the oracle table+hyperlink to HYBRIDER, the performance can reach 44% EM. These results indicate plenty of room for the fusion model and retriever model to improve in future research.
> >
> > “Boil down to retrieve table and then associated documents”:
> > Our best iterative system considers both information sources in two rounds based on the match scores. **If we fix the iterative retriever to first retrieve table rows and then retrieve linked documents, this setting leads to much lower top-K recall (26% F1), 11% worse than the one which retrieves both table+text in both two rounds (37% F1 in Figure 7)**. This reveals the importance of considering diverse retrieval paths of table->text and text->table, a good example is provided in the previous response.
> >
> > “Pre-training”:
> > the results without pre-training are given in the new version, the QA performance drops by 3.5% EM, please see Table 1 (w/o ICT).
> >
> > “Results using just tables and just documents.”:
> > table-only and text-only results are added to Table 1, both of these two cases achieve very low scores of 4-8% EM.
> >
> > “Without Hyperlinks”:
> > We conducted an ablation study and demonstrated our results in Table 1, where we simply use BM25 plus rules to retrieve passages, which leads to a roughly 5-6% EM drop.
> >
> > “Over-complicated, artificial questions”:
> > we use the following rules to decide whether a question is unnatural: 1) questions have two sub-clauses like “what is the birthday of the person who …. in the event that happens …. ”. We modify these questions to contain a single sub-clause. 2) questions that contain very long repetitive text in the document, normally longer than 6 words. We rephrase them to decrease the overly-strong overlap.
> >
> > “Why removing hyperlinks”:
> > The necessity to remove hyperlinks are justified in many recent papers (https://arxiv.org/pdf/2010.12527.pdf, https://arxiv.org/abs/2009.12756). The basic reason to prevent models from over-exploitation of hyperlink annotations which is unique in Wikipedia. The design can make the models generalizable to broader domains.
> >
> > “Multihop retrieval artifacts of splitting the table”:
> > The multi-hop retrieval is not an artifact of table segmentation. The bridging questions only requires one table segment, the comparison questions can be answered by retrieving two independent table segments simultaneously, the argmax/argmin questions can exploit the special [MAX], [MIN] tokens to retrieve the table segment. **Thus, a table segmentation already captures the sufficient information for answering a question**. It’s not the cause for multi-hop retrieval. **Multi-round retrieval is caused by the fact that the information for answering a question is distributed in table and text data**. As their dependency (hyperlinks) is not given in advance, we need to first retrieve one side and then decide what to retrieve on the other side in the second/third round. In other words, after the table segment/passage is retrieved, we do *not* have the hyperlinks to the passages during inference. Therefore, we will have to consider all 5M passages/table segments as the context if we do not do another round of retrieval.
> > Also, note that retrieving the text based on tables is still a very difficult task, see Fig 5 for details.
> >
> >
> > Suggested Improvements:
> > Dual-encode notation: changed in the new version.
> > Lebron James Table Example: changed in the new version.
> > Sparse Fusion retriever: added to Appendix B.3.
> > Typos: fixed in the new version.

---

### Author Response · Authors · 2020-11-24
**General Response and Change Log**

We thank all the reviewers again for their acknowledgment of our paper, and their valuable feedback. We have updated the paper to address the reviewers' concerns. We summarize the changes we made to the paper as follows:

- For Baseline, we add HYBRIDER + Top-K, Text-only, and Table-only baselines.
- For the ablation study
1) we use "Groundtruth Hyperlinks" and "Groundtruth Table + Hyperlinks" to see the potential upper bound of the task.
2) we remove "ICT" and "GPT-2 Query Augmentation" to see their impacts on final EM scores.
3) we investigate standalone entity-linking accuracy of BM25, Dual-Encoder, and GPT-2 to investigate their linking performance.
- For the error analysis
1) we add breakdown analysis for 1-hop, 2-hop, 3-hop questions.
2) we attribute the retriever errors into four typical categories and conduct error analysis for these fine-grained categories.
- For sensitivity analysis:
1) we gradually increase the length limit @K to see the retriever's performance curve.
2) we gradually increase the length limit @K to see the reader's performance curve.

We also polished the paper to improve its clarity. We hope our responses could address the reviewers' concerns, and we are happy to answer any question if possible.

---

### Author Response · Authors · 2021-02-10
**Camera-Ready Version Update**

Thank AC and anonymous reviewers for the valuable comments and constructive suggestions. We revised the paper based on the feedback by AC/reviewers and simplifies the notations used in the paper.

1. We provide more ablation results to the Appendix.
2. We tone down our claims about the naturalness of the dataset in the introduction and stress the questions collected are not as natural as NQ.
3. We improved the figures in the paper and add more visual notations to help the readers understand the idea of fusion-retriever and cross-block reader.

Feel free to raise any questions or comments and we are happy to address them.

---

### Decision · Program_Chairs · 2021-01-07
**Final Decision**

**Decision:**

Accept (Poster)

**Comment:**

This paper presents a new dataset for open domain QA where the evidence required for answering a question is gathered from both structured data as well as unstructured data. The authors first show that a standard iterative retriever with a BERT based reader performs poorly on this task. They then propose fused retrieval (grouping relevant tabular and textual elements) followed by a cross-block reader which improves performance.

R4 has raised strong objections about the artificiality of the dataset. I agree with that and it is unfortunate that the authors did not adequately address the reviewer's concern but instead digressed a bit. As suggested by R4, the authors should tone down their claims about the nature of the dataset. The authors should also simplify the presentation of the dataset as suggested by R2 and not make it unnecessarily complex for the reader.

However, overall, based on reviewer feedback, the authors have made significant changes to the paper. In particular they have added more baselines, ablation studies and error analysis which makes the paper much more informative.

I am okay with this paper getting accepted with the assumption that the authors will make the changes suggested above.